# Production of Secondary Metabolites from Cell Cultures of *Sageretia thea* (Osbeck) M.C. Johnst. Using Balloon-Type Bubble Bioreactors

**DOI:** 10.3390/plants12061390

**Published:** 2023-03-21

**Authors:** Ji-Hye Kim, Jong-Eun Han, Hosakatte Niranjana Murthy, Ja-Young Kim, Mi-Jin Kim, Taek-Kyu Jeong, So-Young Park

**Affiliations:** 1Department of Horticultural Science, Chungbuk National University, Cheongju-si 28644, Republic of Korea; 2Department of Botany, Karnatak University, Dharwad 580003, India; 3Saimdang Cosmetics Co., Ltd., 143, Yangcheongsongdae-gil, Ochang-eup, Cheongwon-gu, Cheongju-si 28118, Republic of Koreajungtk@saimdang.co.kr (T.-K.J.)

**Keywords:** antioxidant activity, bioreactors, catechin, cell suspension cultures, chlorogenic acid, elicitation, phenolics, flavonoids

## Abstract

*Sageretia thea* is used in the preparation of herbal medicine in China and Korea; this plant is rich in various bioactive compounds, including phenolics and flavonoids. The objective of the current study was to enhance the production of phenolic compounds in plant cell suspension cultures of *Sageretia thea*. Optimum callus was induced from cotyledon explants on MS medium containing 2,4-dichlorophenoxyacetic acid (2,4-D; 0.5 mg L^−1^), naphthalene acetic acid (NAA, 0.5 mg L^−1^), kinetin (KN; 0.1 mg L^−1^) and sucrose (30 g L^−1^). Browning of callus was successfully avoided by using 200 mg L^−1^ ascorbic acid in the callus cultures. The elicitor effect of methyl jasmonate (MeJA), salicylic acid (SA), and sodium nitroprusside (SNP) was studied in cell suspension cultures, and the addition of 200 µM MeJA was found suitable for elicitation of phenolic accumulation in the cultured cells. Phenolic and flavonoid content and antioxidant activity were determined using 2,2 Diphenyl 1 picrylhydrazyl (DPPH), 2,2′-azino-bis (3-ethybenzothiazoline-6-sulphonic acid (ABTS), ferric reducing antioxidant power (FRAP) assays and results showed that cell cultures possessed highest phenolic and flavonoid content as well as highest DPPH, ABTS, and FRAP activities. Cell suspension cultures were established using 5 L capacity balloon-type bubble bioreactors using 2 L of MS medium 30 g L^−1^ sucrose and 0.5 mg L^−1^ 2,4-D, 0.5 mg L^−1^ NAA, and 0.1 mg L^−1^ KN. The optimum yield of 230.81 g of fresh biomass and 16.48 g of dry biomass was evident after four weeks of cultures. High-pressure liquid chromatography (HPLC) analysis showed the cell biomass produced in bioreactors possessed higher concentrations of catechin hydrate, chlorogenic acid, naringenin, and other phenolic compounds.

## 1. Introduction

*Sageretia thea* (Osbeck) M.C. Johnst. belong to the family Rhamnaceae is an evergreen shrub that grows up to 3 m high; spines few to numerous; branchlets glabrous to pubescent. The native range of this species is Eritrea to North Somalia, Arabian Peninsula, Central and South China to Peninsula Malaysia, and Temperate East Asia (Korea and Japan). It is a scrambling shrub and grows primarily in the subtropical biome [1]. It is commonly called ‘Sweet plum’ or ‘Chinese sweet plum’ since it bears plum-like fruits that are edible and possess’ higher nutritional value [2]. Fruits are rich in minerals, organic acids, and fatty acids. They are also abundant with phenolics, flavonoids, and anthocyanins [2]. *S. thea* plants are usually used in the preparation of bonsai plants that are having a very high ornamental value in China, Korea, and Japan. The leaves are used in the preparation of green tea. Traditionally *S. thea* is used in the preparation of herbal medicine for the treatment of hepatitis and fever in China and Korea [3,4,5]. Various bioactive compounds such as friedelin, synergic acid, betasitosterol, daucosterol, gluco-synrigic acid, and taraxerol were isolated from the aerial parts of *S. thea* [6]. Several flavanol glycosides, namely, 7-O-methylmearnsitrin, myricentrin, kaempferol 3-O-R-L-rhamonopyranoside, europetin 3-O-R-L-rhamonisde, and 7-O-methyl quercetin 3-O-R-L-rhamnopyranoside have been sequestered from leaves of *S. thea* [7]. Myrictrin and 7-O-methylmearnsitrin have demonstrated a strong in vitro antioxidant activity higher than α-tocopherol [7]. Besides, flavonoid-rich fractions obtained from leaves *S. thea* have shown a protective effect on low-density lipoprotein against oxidative modifications [3]. Pharmaceutical analysis of the fruits of *S. thea* has exhibited antioxidant, anti-diabetic, and anti-melanogenesis activities [2,8]. The leaf extracts have displayed the inhibition of oxidation of low-density lipoprotein through their antioxidant and HIV type 1 protease activities [2]. Furthermore, leaf extracts have demonstrated anticancer activities against human breast cancer (MDA-MB-231) and colorectal cancer cells (SW 480) [9,10]. In view of the above, *S. thea* raw material has been used in the pharmaceutical and cosmetic industries.

There is a tremendous demand for *S. thea* raw material by the pharmaceutical and cosmetic industries in Korea and Japan. Recently, plant cell and organ cultures have emerged as alternatives for the production of plant-based raw materials for the cosmetic and food industries [11,12]. Plant cell and organ cultures are advantageous options for the production of bioactive compounds for biomedical and cosmetic purposes because they represent standardized, contaminant-free, and bio-sustainable systems [13,14,15,16]. Cell and organ culture system has been successfully established in many plants such as *Panax ginseng* [17,18], *Echinacea* species [19], *Eleutherococcus* species [20], *Hypericum peforatum* [21], *Dendrobium candidum* [22,23] and biomass produced in bioreactor system is successfully used in nutraceutical and cosmetic industries. The hypothesis to be tested in this study is the verify the effect of growth regulators, and elicitors on cell suspension cultures of *S. thea*, on biomass and secondary metabolites production. Furthermore, to verify the possibilities of production of *S. thea* on biomass and secondary metabolites in bioreactor cultures. In the current study, we are aiming at the production of phenolic compounds from the cell culture of *S. thea*, we established a cell culture system in flasks and balloon-type bioreactor cultures. We also adopted elicitation technology for the hyperproduction of phenolics in cell cultures of *S. thea*.

## 2. Results

### 2.1. The Effect of Growth Regulators and Explant Type on the Callus Induction

The results of the effect of growth regulators and explant type on callus induction of *S. thea* are presented in Figure 1. MS medium supplemented with 2,4-D at 1 mg L^−1^ triggered 31.7%, 46.7%, 56.7%, and 20% callus induction from complete seed, seed halves, cotyledon, and leaf explants respectively. Addition of BA (0.1 mg L^−1^) or KN (0.1 mg L^−1^) to the 2,4-D containing medium or combination of 2,4-D (1 mg L^−1^), NAA (0.1 mg L^−1^), BA (0.1 mg L^−1^) or 2,4-D (0.5 mg L^−1^), NAA (0.5 mg L^−1^), BA (0.1 mg L^−1^) has not triggered a further improvement in the induction of callus (Figure 1). However, a combination of 2,4-D (1.0 mg L^−1^), NAA (0.1 mg L^−1^), KN (0.1 mg L^−1^), and 2,4-D (0.5 mg L^−1^), NAA (0.5 mg L^−1^), BA (0.1 mg L^−1^) has involved in the induction of higher amount of callus from various explants of *S. thea* (Figure 1). On medium supplemented 2,4-D (1.0 mg L^−1^), NAA (0.1 mg L^−1^), KN (0.1 mg L^−1^) 23.3%, 80.0%, 96.7%, and 26.7% callus was induced from seed, seed halves, cotyledon, and leaf explants, respectively (Figure 1), however, along with callus adventitious root regeneration was recorded on this medium. Whereas, on MS medium supplemented with 2,4-D (0.5 mg L^−1^), NAA (0.5 mg L^−1^), BA (0.1 mg L^−1^) seed, seed halves, cotyledon, and leaf explants have developed 15.0%, 43.3%, 93.3%, and 36.7% callus respectively and there was no adventitious roots development with such callus (Figure 1 and Figure 2). Of the four explants used (seed, seed halves, cotyledons, and leaf explants), auxins and cytokinins and explants vs. culture medium showed statistical significance (F value 13.83, degree of freedom 27, and *p*-value ≤ 0.05) (Figure 1). Seed-derived callus was friable and light yellow (Figure 2), whereas the callus induced from cotyledon, both compact and friable, subsequently turned creamy in color. Callus derived from leaf explants was brown in callus; however, it did not survive in subsequent subcultures (Figure 2).

### 2.2. Selection of Friable Callus

Histological analysis of calli-derived from seeds, cotyledons, and leaves was of the following five different types: white, hard, soft, yellowish, and brown (Figure 3). Among these types, soft and yellowish callus possessed actively dividing cells with intercellular spaces and prominent nuclei. Whereas white, hard, and brown calli possess compactly arranged cells and had large vacuoles (Figure 3). In addition, this callus showed starch granules and thick cell walls and was not involved in cell division. Thus, the soft and yellowish callus bore rapidly dividing cells, which are composed of loosely arranged, undifferentiated cells.

### 2.3. Effect of Antioxidants on Callus Proliferation and Overcoming the Problems of Browning of the Medium

The callus, which was friable and actively growing, were sub-cultured on MS medium supplemented with 2,4-D (0.5 mg L^−1^), NAA (0.5 mg L^−1^), and KN (0.1 mg L^−1^) for the proliferation of callus. However, the accumulation of phenolics and medium browning was a problem in subsequent cultures. Therefore, in the current study, callus of *S. thea* was sub-cultured on a medium containing 0, 100, 200 mg L^−1^ ascorbic acid (ASA) or 0, 10, 20 mg L^−1^ citric acid (CA) or 0, 10, 20 mg L^−1^ polyvinyl pyrrolidone (PVP) to overcome the problems of browning of callus and results are presented in Figure 4. The callus extract of brown callus showed an absorbance of 0.19 with a spectrophotometer at 420 nm. In contrast, the callus treated with 200 mg L^−1^ ASA did not involve browning, and such callus extract showed an optical absorbance of 0.01 at 420 nm (F value 3.83, degree of freedom 6, and *p*-value ≤ 0.05) (Figure 4A). Furthermore, the callus treated with 200 mg L^−1^ ASA was involved in rapid growth and attained a biomass of 1.04 g fresh weight per callus mass (F value 3.83, degree of freedom 6, and *p*-value ≤ 0.05) (Figure 4B). Whereas the other treatments, such as treatment with CA and PVP, were not beneficial, and they could not control the browning of callus completely (Figure 4C).

### 2.4. Establishment of Suspension Cultures in Erlenmeyer Flasks and Elicitation

Erlenmeyer’s flasks were used for the initial cell suspension research. *S. thea* cell suspension cultures that were cultured using MS liquid medium supplemented with 2,4-D (0.5 mg L^−1^), NAA (0.5 mg L^−1^), and KN (0.1 mg L^−1^) showed an initial lag phase for seven days (one week) and exponential phase from 7 to 21 days (up to three weeks) in culture, after that cell entered into stationary phase. After two weeks of culture initiation, elicitors including sodium nitroprusside (SNP; nitric oxide producer), methyl jasmonate (MeJA), and salicylic acid (SA) were added to cell suspension cultures of *S. thea* to test the concentration and effectiveness of the elicitor on the biosynthesis of phenolic compounds. The results of elicitor treatment depicted that the growth and accumulation of biomass were inhibited with the addition of elicitors SNP and MeJA (Figure 5A,B). MeJA specifically inhibited the accumulation of biomass with the increment in the concentration of MeJA (F value 3.83, degree of freedom 6, and *p*-value ≤ 0.05) (Figure 5A). The biomass accumulated in the control was 21.4 g after four weeks of culture, and it was reduced to 15.1 g, 13.5 g, and 12.5 g with the addition of 50, 100, and 200 µM, respectively. A similar decrease in biomass content was also noticed with SA treatments. Nevertheless, there was an increment accumulation of total phenolics and total flavonoids that was evident with MeJA treatments (Figure 6). Of the various elicitors tested, MeJA was efficient in triggering the accumulation of phenolic and flavonoid contents in the cultured cells of *S. thea* (Figure 6). The amount of phenolic content was 5.9 mg g^−1^ GAE with control cultures, whereas, in MeJA-treated cell cultures, the concentration of total phenolics was 37.5, 36.9, and 34.5 mg g^−1^ GAE (F value 3.83, degree of freedom 6, and *p*-value ≤ 0.05) (Figure 6). Similarly, the amount of total flavonoids was 1.6 mg g^−1^ CAT equivalents in the control, while the total flavonoid content was 18.4, 17.5, and 15.1 mg g^−1^ CAT equivalents with different concentrations of MeJA treatments (F value 3.83, degree of freedom 6, and *p*-value ≤ 0.05) (Figure 6). We carried out an antioxidant analysis of extracts of *S. thea* cell cultures that were treated with different elicitors, and the results are presented in Figure 7 (F value 3.83, degree of freedom 6, and *p*-value ≤ 0.05). Analysis of DPPH radical scavenging activity (Figure 7A), ABTS radical scavenging activity (Figure 7B), and FRAP assay (Figure 7C) all demonstrated that extracts obtained from MeJA elicited cell cultures possessed the highest activities compared to cell extracts of SA and SNP-treated cultures and even control. The IC_50_ values ABTS scavenging activity were in the range of 0.6–0.8 mg mL^−1^ with the MeJA treated extracts, whereas it was 9.3 mg mL^−1^ with the control cell extracts (Figure 7D). These results demonstrate that MeJA-treated cells are actively involved in secondary metabolism and involved in the enhanced accumulation of bioactive compounds.

### 2.5. Establishment of Bioreactor Cultures for the Production of Biomass and Phenolics

Ten g L^−1^ cells were cultured in bioreactors, cultures were aerated with 0.1 vvm, and treated with 200 µM MeJA after three weeks of culture and maintained for another week. The bioreactor cultures showed a typical lag phase for one week and an exponential phase for up to four weeks. Accumulation of 230.81 g of fresh biomass and 16.48 g of dry biomass was evident after four weeks of culture (F value 714.62, degree of freedom 4, and *p*-value ≤ 0.05) (Figure 8 and Figure 9). The cell growth index was 8.2. These results depict that bioreactor cultures are efficient in the accumulation of biomass.

The findings of an HPLC examination of the phenolics present in the biomass are shown in Figure 10 and Figure 11. Seven major phenolics were recognized in the cell extracts when compared to standards (Figure 10). The major phenolics including catechin hydrate (382.0 µg g^−1^ DW), chlorogenic acid (108.1 µg g^−1^ DW), naringin (82.9 µg g^−1^ DW), apigenin (15.5 µg g^−1^ DW), ruteolin (14.7 µg g^−1^ DW), gallic acid (13.2 µg g^−1^ DW) and ferulic acid (1.5 µg g^−1^ DW) were recorded through HPLC analysis (F value 714.62, degree of freedom 4, and *p*-value ≤ 0.05) (Figure 11). Comparison of cultures without elicitation and with elicitation with MeJA revealed that 28.7, 24, and 4.8-fold increments in accumulation of catechin hydrate, naringin, and chlorogenic acid are evident (Figure 11).

## 3. Discussion

Plant secondary metabolites such as phenolics and flavonoids have very high therapeutic values such as antioxidant, anti-inflammatory, anticancer, antihypertensive, antihyperglycemic, neuroprotective, and hypolipidemic activities [24]. Production of these compounds in natural plants depends upon various factors such as species, populations, and edaphic and environmental conditions where they grow. Variability in the accumulation of bioactive compounds is also evident depending on the phenological stages of development [25]. In contrast production of bioactive compounds through in vitro culture of plant cells and organs has emerged as an attractive alternative. Growth and accumulation of biomass and production of metabolites can be controlled very easily with in vitro conditions. Additionally, in vitro cultivation of plant cells allows for the manipulation of growth variables, and with additions of elicitors and precursors, hyperproduction of valuable bioactive active compounds could be achieved [14,15]. For the production of phenolic compounds; therefore, we established cell suspension cultures of *S. thea* in the current investigation.

Plant growth regulators play an important role in triggering the growth and development of explants cultured in vitro [26]. Specific explant requires a particular auxin or cytokinin or combination of auxin and cytokinin for callus induction, and it also depends on the levels of endogenous hormones present with the explants. Auxins are involved in cell division, tissue differentiation, embryogenesis, and rhizogenesis [27]. In the present study, we tested the effect of 2,4-D (0.5, 1 mg L^−1^), NAA (0.5, 1.0 mg L^−1^), BA (1.0 mg L^−1^), and KN (1 mg L^−1^) individually or in combination. A combination of 2,4-D (0.5 mg L^−1^) with NAA (0.1 or 0.5 mg L^−1^) and KN (0.1 mg L^−1^) has triggered the highest amount of callus induction in seed, cotyledon, and leaf explants of *S. thea*. Similar to the present results addition of NAA along with other growth regulators has induced the highest amount of callus in *Cananga odorata* and *Scropholaria stiata* [28,29]. The selection of suitable explants is also critical in the induction of a greater amount of callus [30]. In the current study highest callus induction was recorded from seed (80%) and cotyledon (96.7%) explants of *S. thea* on the combination of medium containing 2,4-D (1.0 mg L^−1^), NAA (0.1 mg L^−1^) and KN (0.1 mg L^−1^), however, the formation of roots alongside callus was noticed on both explants after four weeks of culture. Similarly, in *Hylocerus costaricensis* NAA induced rhizogenesis along with callus on NAA supplemented medium [31]. However, on the MS medium containing 2,4-D (0.5 mg L^−1^), NAA (0.5 mg L^−1^), and KN (0.1 mg L^−1^), seed and cotyledon explants have produced 43.3% and 93.3% callus formation without the intervention of rhizogenesis. Variations among explant types for callus induction have also been reported in other plant species, such as *Calendula officinalis* and *Chorisia spciosa* [32,33].

The friability of callus tissue is highly desirable for establishing cell suspension culture [34]. Histological analysis is very useful to verify the process friability of the callus. The soft and yellowish callus, among the numerous forms of calli found in the current studies, had cells that were actively proliferating and had intercellular gaps and conspicuous nuclei. While dark, hard, and white calli have big vacuoles and compactly organized cells. This callus also exhibited thick cell walls, starch granules, and a lack of cell division. Similar to the present results, Betekhtin et al. [35] observed characteristic cells that are compact, thick-walled, and accumulation of insoluble starch grains in the non-friable callus. As a result, the loosely packed, undifferentiated cells that make up the rapidly dividing cells in the soft, yellowish callus were selected for further sub-culturing and callus proliferation.

Oxidative browning is a common problem in plant tissue cultures, which results in reduced growth of the cultured explants or callus [36,37]. The underlying phenomenon is the accumulation and subsequent oxidation of phenolic compounds in the tissues. Several methods have been followed to overcome the initial browning of the medium, including the use of antioxidants such as ascorbic acid and citric acid or the use of absorbents such as polyvinyl pyrrolidone and activated charcoal [37,38]. With our results, we realized callus browning during subsequent sub-cultures. To overcome the browning of the callus, we used 100, 200 mg L^−1^ ascorbic acid (ASA) or 0, 10, 20 mg L^−1^ citric acid (CA) or 0, 10, 20 mg L^−1^ polyvinyl pyrrolidone (PVP). Of the varied treatments, cultures treated with 200 mg L^−1^ ASA could be able to control the browning of the callus in subsequent subcultures. Similar to the current results, ASA has been efficiently used in overcoming the browning of callus in tissue cultures of *Themeda qundrivalis* [39], *Musa* species [40], and *Glycyrrhiza glabra* [41].

Elicitation is one of the most effective and widely employed biotechnological tools for enhanced biosynthesis and accumulation of secondary metabolite in plant tissue culture [42]. Elicitors of biotic and abiotic origin can trigger the biosynthesis of specific metabolites in cell and organ cultures [14,43]. The optimization of various parameters, such as elicitor type, concentration, duration of exposure, and treatment schedule is essential for the effectiveness of the elicitation strategies. Salicylic acid (SA), methyl jasmonate (MeJA), jasmonic acid (MJ), and other signaling molecules are frequently utilized as elicitors to stimulate secondary metabolite biosynthesis and enhancement [44,45,46,47]. In our studies, we established cell suspension cultures and tested the effect of MeJA, SA, and sodium nitroprusside (SNP; nitric oxide producer) at 50, 100, and 200 µM concentrations. MeJA was effective in causing the accumulation of phenolic and flavonoid contents in the cultured cells of *S. thea* when compared to other elicitors tested. In total, 5.9 mg g^−1^ GAE of phenols were present in control cultures, but 37.5, 36.9, and 34.5 mg g^−1^ GAE were present in cell cultures that had received MeJA treatment. Similarly, the total flavonoid content was 18.4, 17.5, and 15.1 mg g^−1^ CAT equivalents with various doses of MeJA treatments, compared to 1.6 mg g^−1^ CAT equivalents in the control. Similar to the present findings, MeJA has successfully been employed as an elicitor for the enhanced accumulation of ginsenosides in *Panax ginseng* cell and adventitious root cultures [44,45] as well as the generation of phenolic compounds in *Thevetia peruviana* cell cultures [48]. SA and SNP were not efficient in stimulating the accumulation of secondary metabolites with cell cultures of *S. thea*. However, SA and nitric oxide have been used as potent elicitors in callus/cell cultures of *Hypericum perforatum* [49] and *Catharanthus roseus* [50] for the production of hypericin and catharanthine, respectively.

The production of biomass and secondary metabolites from cell and organ cultures is excellent in bioreactor cultures, and various factors, including medium pH, aeration, and gases, can be controlled in a way that is growth-specific. Cultured cells will effectively utilize nutrients and participate in the creation and development of metabolites [14,15]. Additionally, bioreactor cultures enhance production quality, lower production costs, and allow for a process to scale up [51]. The results of the current study of bioreactor cultures showed a higher accumulation of biomass (230.81 g of fresh biomass and 16.48 g of dry biomass) as well as metabolites. The generation of biomass and secondary metabolites in a variety of plant species, such as *Panax ginseng* [17], *Echinacea* species [19], and *Hypericum perforatum* [21], has also been investigated using balloon-type bubble bioreactors. Comparing cultures with and without elicitation with MeJA, it was discovered that there was a 28.7, 24, and 4.8-fold increase in the accumulation of catechin hydrate, naringin, and chlorogenic acid. Due to its antibacterial, anti-inflammatory, antidiabetic, and anticancer activities, the bioactive molecule catechin has a very high therapeutic value. It can also effectively neutralize free radicals and has significant antioxidant properties [52]. While chlorogenic acid has a variety of physiological benefits, including hepatoprotective, gastrointestinal, renoprotective, neurological, and cardiovascular properties [53]. Naringenin, a flavonoid molecule, has been shown to have anti-inflammatory, antioxidant, antibacterial, antiadipogenic, and cardioprotective properties [54]. The bioactive compounds found in abundance in the *S. thea* cell biomass produced in bioreactors allow for the extraction of these fine chemicals as well as the use of the biomass in the production of nutraceuticals and cosmetics.

## 4. Materials and Methods

### 4.1. Plant Material and Seed Germination

*Sageretia thea* (Osbeck) M.C. Johnst. plants were collected from Dongbaekdongsan, which is located in Seonheul-ri, Jocheon-eub, Bukjeju-gun, Jeju-do, Republic of Korea, and maintained in the experimental garden at Chungbuk National University, Republic of Korea. Fruits were collected from the plants grown in the experimental garden and washed thoroughly in running tap water, and surface sterilized in 70% ethanol for 5 s, then in 20% sodium hypochlorite solution with one drop of Tween-20 for 30 min. Fruits were washed three times in sterile water under the laminar flow cabinet, and then the seeds were separated from the fruits. Seeds were washed in sterile distilled water three times and cultured on Murashige and Skoog (MS) [55] medium supplemented with 1.0 mg L^−1^ gibberellic acid, 0.1 mg L^−1^ kinetin, and 30 g L^−1^ sucrose for germination.

### 4.2. Chemicals and Reagents

Methyl jasmoante (MeJA), salicylic acid (SA), sodium nitroprusside (SNP), gallic acid (GA), catechin, Folin-Ciocalteu (FC) reagent, 2,2 Diphenyl 1 picrylhydrazyl (DPPH), 2,2′-azino-bis (3-ethybenzothiazoline-6-sulphonic acid (ABTS), 2,4,6-tripyridyl-s-triazine (TPTZ), ascorbic acid (ASA), citric acid (CA), polyvinylpyrrolidone (PVP) were procured from Sigma-Aldrich chemicals (St. Louis, MO, USA). All the tissue culture chemicals, growth regulators such as 2,4-dichlorophenoxyacetic acid (2,4-D), naphthalene acetic acid (NAA), 6-benzyladenine (BA), kinetin (KN), gibberellic acid (GA_3_), gelrite, were obtained from Ducefa Biochemie, Haarlem, The Netherlands. High-pressure liquid chromatography (HPLC) standard phenolic compounds were procured from ChromaDex, Longmont, CO, USA.

### 4.3. Callus Induction

Entire seeds, seeds divided into two halves, cotyledons, and leaves obtained from young plantlets were used for callus induction. Explants were cultured on MS medium containing 30 g L^−1^ sucrose, 2,4-D (0.5, 1 mg L^−1^), NAA (0.5, 1.0 mg L^−1^), BA (1.0 mg L^−1^), KN (1 mg L^−1^) individually or in combination. Medium pH was adjusted to 5.8 and then added with 2.4 g L^−1^ gelrite and autoclaved at 121 °C and 121 Kilopascals for 15 min. The cultures were maintained in dark at 25 ± 1 °C for 4 weeks in culture rooms.

### 4.4. Histological Analysis of Callus

For the histological analysis of callus, each type of callus (hard, soft, callus of different colors such as white, yellowish, brown, irrespective of their origin from seeds, cotyledons, and leaves) 0.5 mm^3^ callus was fixed in formalin, glacial acetic acid, and 95% ethyl alcohol and water (10: 5: 50: 35) solution for two days. Then callus was degassed and dehydrated with an alcohol series. After infiltration with Technovit 7100 (Kulzer Technik, Wertheim, Germany), a mold was made by polymerizing with an embedding solution. Microtome (Leica, Nussloch, Germany) sections (5 µm) were taken and attached to glass slides and stained with periodic acid Schiff reagent and toluidine blue O. The preparations were observed using an optical microscope (Leica, Germany).

### 4.5. Raising of Callus Cultures with Supplementation of Antioxidants

The callus developed from the different explants invariably use to accumulate phenolic accumulation that hindered the callus growth. To overcome initial phenolic accumulation and to facilitate the growth of the callus, callus cultures sub-cultured MS medium with growth regulators 0.5 mg L^−1^ 2,4-D, 0.5 mg L^−1^ NAA, and 0.1 mg L^−1^ KN and various antioxidants such as 100, 200 mg L^−1^ ASA or 10, 20 mg L^−1^ citric acid or 10, 20 mg L^−1^ PVP. After 4 weeks of culture, callus growth and browning of cultures with phenolics were investigated.

### 4.6. Establishment of Cell Suspension Culture in Erlenmeyer’s Flasks and Elicitation of Cultures

*S. thea* cell suspensions were established in 100 mL Erlenmeyer’s flasks with 50 mL of MS liquid medium, 30 g L^−1^ of sucrose, 0.5 mg L^−1^ of 2,4-D, 0.5 mg L^−1^ of NAA, and 0.1 mg L^−1^ of KN. In total, 3 g of friable callus was added to 50 mL of liquid medium, and the cultures were maintained on an orbital shaker at 120 rpm for four weeks. The cultures were added with elicitors such as salicylic acid (SA), methyl jasmonate (MeJA), or sodium nitroprusside (SNP) at a concentration of 50, 100, and 200 µM after two weeks of culture and maintained for another two weeks.

### 4.7. Establishment of Cell Suspension Cultures in Balloon-Type Bubble Bioreactors

For the production of *S. thea* biomass, cell suspension cultures were established in 5 L balloon-type bubble bioreactors (Samsung Biotech, Seoul, Republic of Korea) containing 2 L of MS liquid medium containing 30 g L^−1^ sucrose and 0.5 mg L^−1^ 2,4-D, 0.5 mg L^−1^ NAA and 0.1 mg L^−1^ KN. In total, 20 g of cells were inoculated to the medium. After three weeks of culture initiation, 100 µM MeJA was added to elicit the cells to involve in secondary metabolism and maintained for another week. The bioreactor cultures were maintained in a culture room, and the dark were aerated with sterile air at 0.1 vvm (air volume/medium volume/min). After four weeks of culture, cells were harvested by bypassing the medium through a stainless-steel sieve, and cells were washed thoroughly with sterile distilled water. The fresh weight (FW) was determined after air drying cells, dry weight of the cell was determined by drying the cell biomass in a freeze-dryer at −80 °C for three days. The growth index (Gi) was calculated on the basis of weight of dry tissue according to the following formula: Gi = (DW1 − DW0)/DW0: where DW0 was the weight of inoculum and DW1 was the final weight of tissue after a culture growth period. The total phenolic content (TPC), total flavonoid content (TFC), and antioxidant analysis were all calculated using the cell biomass.

### 4.8. Preparation of Cell Extracts

The freeze-dried cells (0.1 g) were taken in 10 mL of ethanol and subjected to ultrasonic waves at 36 °C for 1 h. The ethanol extract was filtered through Whatman (grade 2) filter paper, and the filtrate was used for the analysis of phenolics, flavonoids, and antioxidant activities.

### 4.9. Estimation of Total Phenolic Content

Total phenolic content (TPC) was estimated by using the Folin Ciocalteu reagent method, as described by Murthy et al. [56], with some modifications. Briefly, a known amount of sample was taken and made up to 3 mL with distilled water, and 0.1 mL of 2 N Folin Ciocalteu reagent was added, followed by incubation for 6 min, and then 0.5 mL of 20% Na_2_CO_3_ was added to each tube. Tubes were kept in warm water for 30 min, and the absorbance was read at 760 nm using a UV-Visible spectrophotometer. Gallic acid was used as the standard compound.

### 4.10. Estimation of Total Flavonoid Content

The flavonoid content of extracts was analyzed as described by Pekal and Pyrzynskaet [57]. To brief, 0.1 mL of extract was taken and made up the volume to 3 mL by using distilled water, followed by the addition of 0.15 mL of 10% AlCl_3_ and 2 mL of 1 M NaOH after 5 min of incubation at room temperature. Solutions were vortexed, and absorbance was measured at 510 nm. Catechin was used as standard.

### 4.11. Analysis of Antioxidant Activities

#### 4.11.1. 2,2 Diphenyl 1 picrylhydrazyl (DPPH) Radical Scavenging Assay

Extract (0.1 mL) was added with 1.9 mL of 0.1 mM DPPH solution prepared in ethanol. The tubes were vortexed and incubated in the dark for 15 min. The discoloration of the DPPH solution was measured at 517 nm against ethanol as blank using a UV-visible spectrophotometer. Gallic acid was used as standard, and the activity of the extracts was expressed as mg gallic acid equivalent (GAE)/g extract [58].

#### 4.11.2. 2,2′-Azino-bis (3-ethybenzothiazoline-6-sulphonic Acid (ABTS) Assay

The ABTS assay was carried out as per the method of Re [59]. The ABTS solution was prepared by mixing 7 mM of ABTS and 2.45 mM potassium persulfate in a ratio of 1:1 and stored in the dark for 24 h. At the time of analysis, the ABTS solution was diluted with phosphate buffer (pH 7.3) to obtain the value of 0.70 at 732 nm. Fifty microliters of the extract were added to 950 microliters of diluted ABTS solution, and the mixture was allowed to stay in the dark for 10 min then absorbance was measured at 732 nm using UV-visible spectrophotometry. Antioxidant activity was expressed in percentage, i.e., ABTS radical scavenging activity = absorbance of control solution-absorbance of sample solution/absorbance of control solution × 100.

#### 4.11.3. Ferric Reducing Antioxidant Power (FRAP) Assay

FRAP assay was carried out according to the method described by Benzie and Strain [60]. FRAP reagent was prepared by mixing 300 mM acetate buffer of pH 3.6, 10 mM 2,4,6-tripyridyl-s-triazine (TPTZ) in 40 mM HCl and 20 mM FeCl_2_.6H_2_O in the ratio 10:1:1. In total, 0.2 mL of extract was added with 3 mL of FRAP reagent, tubes were vortexed and incubated for 6 min at room temperature, and absorbance was measured at 593 nm using a UV-visible spectrophotometer. Ascorbic acid was used as standard, and activity is expressed as mg ascorbic acid equivalent (AAE)/g extract.

### 4.12. Quantification of Phenolic Compounds Using High-Performance Liquid Chromatography (HPLC)

The cell biomass obtained from the cultures was ground in a sterilized mortar. The powdered sample (0.1 g) was mixed with 10 mL of 80% ethanol, and the extract was obtained by ultrasonication, as explained above. The extract was concentrated using nitrogen gas and dissolved in 0.5 mL 80% ethanol and used for analysis. The extract was filtered through a membrane filter (0.45 µm) and used for analysis. HPLC equipment (2690 Separation Module, Waters Chromatography, Milford, CT, USA) included a photodiode array detector (PDA), and compound separation was performed using a Fortis C18 column (5 µm, 150 × 4.6 mm). The mobile phase consisted of acetic acid and water (1:99 *v*/*v*) (solvent A) and acetic acid and acetonitrile (1:99 *v*/*v*) (solvent B) and was filtered using Whatman Glass microfiber filters before use. The flow rate was 1.0 mL.min-1, and the column temperature was 25 °C. The peaks were detected at 280 nm, and compounds were identified and quantified based on the retention time of standards and peak areas.

### 4.13. Statistical Analysis

The results are presented as mean values and standard errors. One-way analysis of variance (ANOVA) was used to determine whether the groups differed significantly. Statistical assessments of the difference between mean values were then assessed using Tukey’s test. A value of *p* = 0.05 was considered to indicate statistically significant differences. All the data were analyzed using a SAS program (Software Version 9.4; SAS Institute, Cary, NC, USA).

## 5. Conclusions

Plant cell culture techniques can be used as alternatives for the production of biomass that are abundant in bioactive materials. In the present study, we established stepwise protocols for the induction of callus and establishment of suspension cultures for the production of phenolic compounds in *S. thea*. Methyl jasmonate elicited cell cultures of *S. thea*, has the highest content of total phenolics and flavonoids, and also demonstrated increased antioxidant activity. The cell biomass was also rich in specific bioactive compounds, including catechin hydrate, chlorogenic acid, naringenin, and others.

## Figures and Tables

**Figure 1 plants-12-01390-f001:**
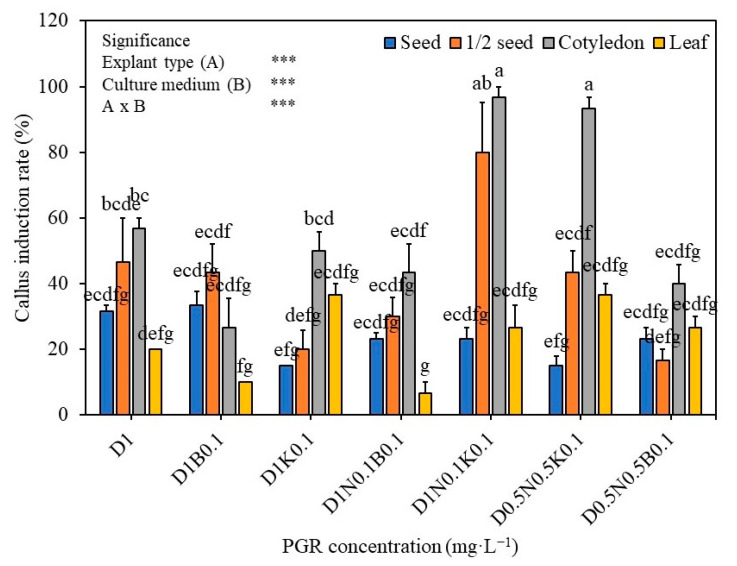
Effect of plant growth regulators on callus induction seed, half seed, cotyledon, and leaf explants of *S. thea* on MS medium containing 2,4-D, NAA, BA, and KN after 4 weeks of culture. Mean values with different alphabetical letters denote significant differences between the values according to Tukey’s test at *p* ≤ 0.05. *** Explants (A) and culture medium (B) effects are statistically significant at *p* ≤ 0.05.

**Figure 2 plants-12-01390-f002:**
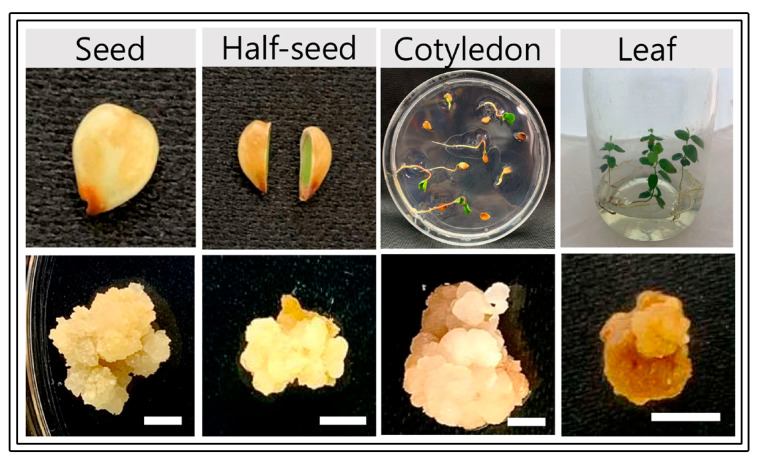
Callus induced from seed, half seed, cotyledon, and leaf explants of *S. thea* on MS medium containing 2,4-D, NAA, BA, and KN after 4 weeks of culture. Scale bars = 1 cm.

**Figure 3 plants-12-01390-f003:**
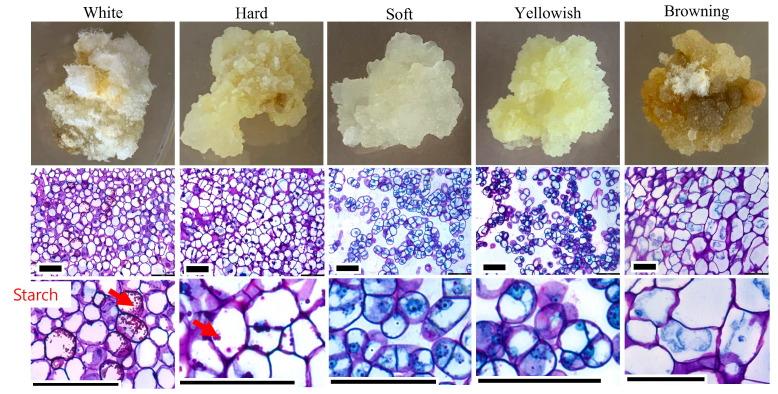
Morphological and histological features of callus regenerated from different explants of *S. thea*. Scale bar = 100 µm. Red arrows indicate starch grains.

**Figure 4 plants-12-01390-f004:**
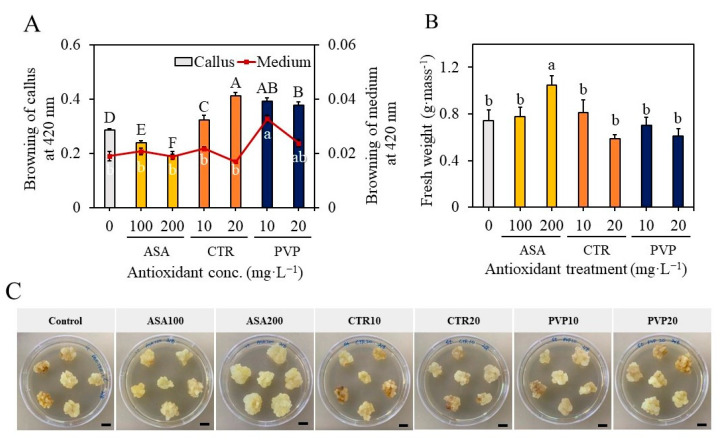
Morphological features of callus after treatment with 100, 200 mg L^−1^ ascorbic acid (ASA) or 10, 20 mg L^−1^ citric acid (CTR) or 10, 20 mg L^−1^ polyvinyl pyrrolidone (PVP). (**A**) Shows the absorbance of cell extract at 420 nm on a spectrophotometer, (**B**) shows the fresh weight of the callus, and (**C**) the morphology of the callus with different treatments. Different letters indicate mean values which are significantly different at *p* ≤ 0.05 according to Tukey’s test.

**Figure 5 plants-12-01390-f005:**
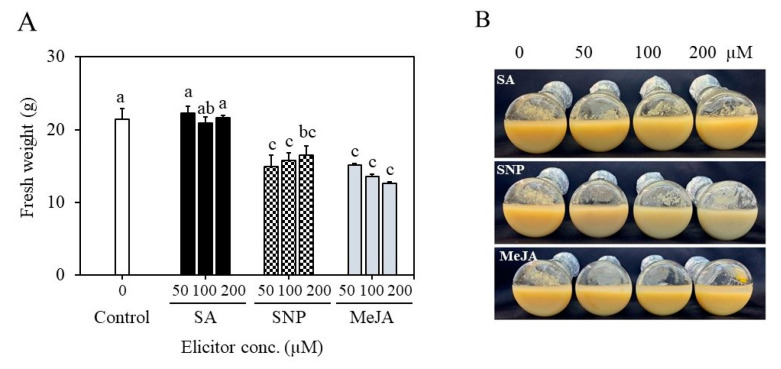
Effect of elicitor treatments on biomass accumulation in cell suspension cultures (**A**). Biomass growth with the increment of elicitor concentrations compared to control (**B**). SA—salicylic acid, SNP—sodium nitroprusside, MeJA—methyl jasmonate. Different letters indicate mean values which are significantly different at *p* ≤ 0.05 according to Tukey’s test.

**Figure 6 plants-12-01390-f006:**
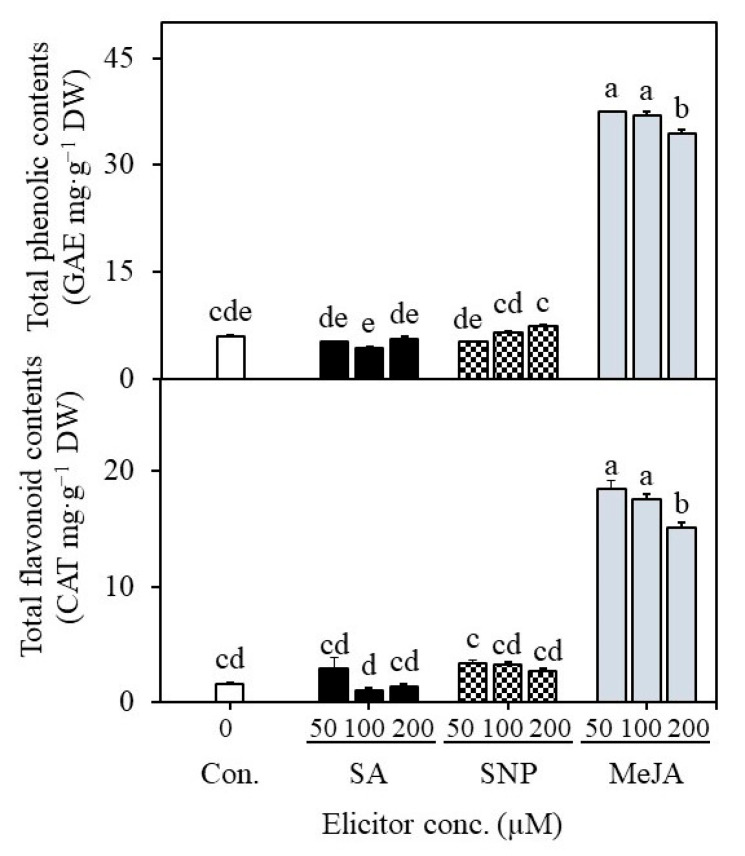
Effect of elicitor treatments on the accumulation of total phenolics and flavonoids in cell suspension cultures. SA—salicylic acid, SNP—sodium nitroprusside, MeJA—methyl jasmonate. Different letters indicate mean values which are significantly different at *p* ≤ 0.05 according to Tukey’s test.

**Figure 7 plants-12-01390-f007:**
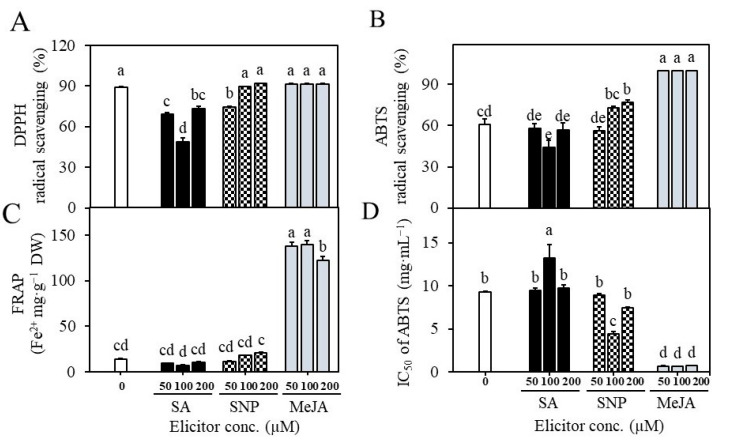
Effect of elicitor treatments on antioxidant activity in cell suspension cultures. SA—salicylic acid, SNP—sodium nitroprusside, MeJA—methyl jasmonate. Effect of elicitors on DPPH radical scavenging activity (**A**), ABTS radical scavenging activity (**B**), FRAP assay (**C**), and IC_50_ values of ABTS activity (**D**). Different letters indicate mean values which are significantly different at *p* ≤ 0.05 according to Tukey’s test.

**Figure 8 plants-12-01390-f008:**
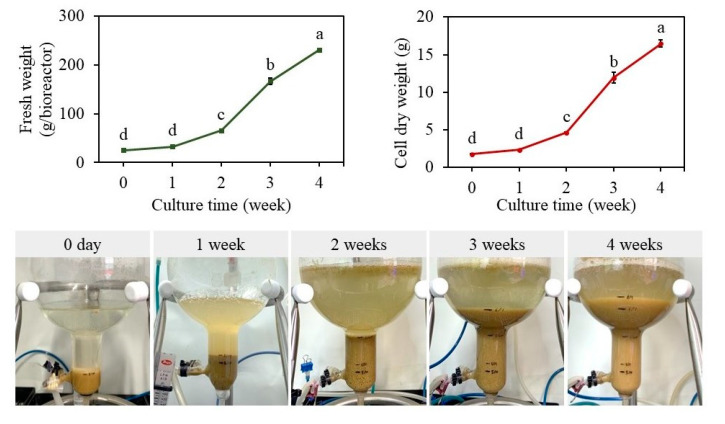
Cell suspension cultures in balloon-type bubble bioreactors: accumulation of fresh biomass, dry biomass over four weeks of culture Different letters indicate mean values which are significantly different at *p* ≤ 0.05 according to Tukey’s test.

**Figure 9 plants-12-01390-f009:**
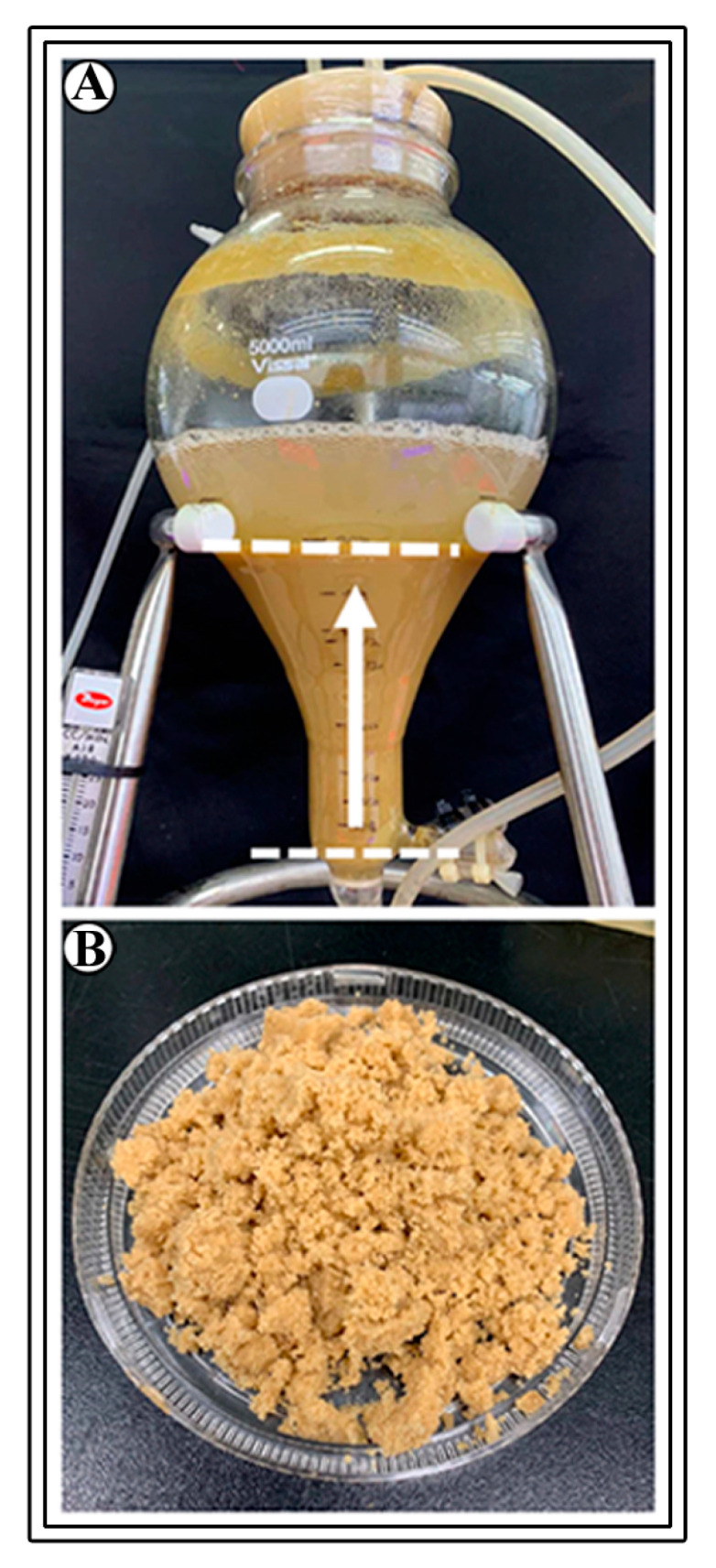
Cell suspension cultures in balloon-type bubble bioreactor. (**A**) Biomass accumulated after 4 weeks of culture, (**B**) Harvested cell biomass.

**Figure 10 plants-12-01390-f010:**
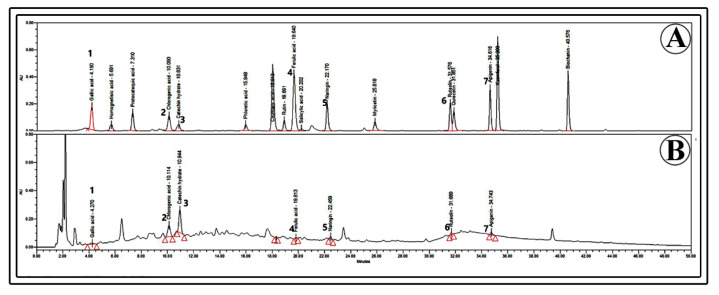
HPLC chromatograms of phenolic compounds. (**A**) Standards, (**B**) gallic acid (1), chlorogenic acid (2), catechin hydrate (3), ferulic acid (4), naringin (5), luteolin (6), and apigenin (7).

**Figure 11 plants-12-01390-f011:**
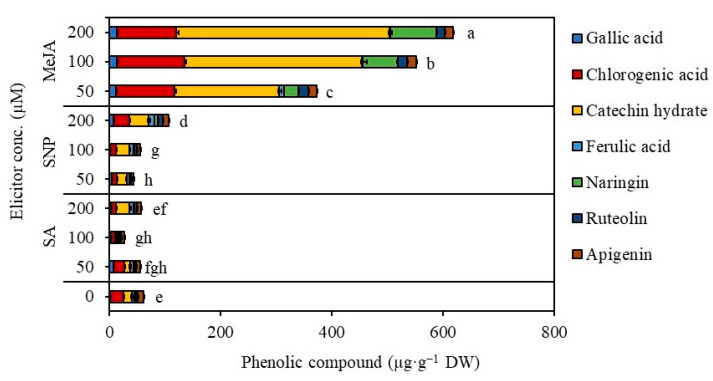
Quantity of phenolic compounds accumulated in cell suspension cultures with elicitor treatments. SA—salicylic acid, SNP—sodium nitroprusside, MeJA—methyl jasmonate. Different letters indicate mean values which are significantly different at *p* ≤ 0.05 according to Tukey’s test.

## Data Availability

The datasets used and/or analyzed during this study are available from the corresponding author upon reasonable request.

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
