# Peer review of "Production of Secondary Metabolites from Cell Cultures of Sageretia thea (Osbeck) M.C. Johnst. Using Balloon-Type Bubble Bioreactors"

_plants, 2023, doi:10.3390/plants12061390_

Round 1
Reviewer 1 Report
These are my main comments on the manuscript (plants-2250568) entitled “Production of secondary metabolites from cell cultures of Sageretia thea (Osbeck) M.C. Johnst. using balloon-type bubble bioreactors”. This work investigates the production of phenolic compounds in plant cell suspension cultures of Sageretia thea. Following substantial revisions should be incorporated in the manuscript prior to acceptance.
1. I have concerns about the manuscript sections that I believe need to be addressed in order to improve its clarity.
2. A hypothesis for this study is needed.
3. In results section, statistical data are needed. Please, provide the F value, degree freedom, and P-value obtained from analysis of variance.
4. Some statistical analyses are missing. Please, revise the results obtained in figures 8 and 11.
5. Other revisions could be checked in PDF attached.

Author Response
Response to reviewers’ comments
Manuscript ID: plants-2250568
Authors are thankful to anonymous reviewers for their valuable comments on the manuscript. We have revised the manuscript in the light of reviewer’s comments and incorporated all the suggestion made by them. All corrections made in the revised manuscript have been done with track changes. Following are the specific changes made in the manuscript.
Reviewer #1
- I have concerns about the manuscript sections that I believe need to be addressed in order to improve its clarity.
Answer: All the suggested corrections are incorporated.
- A hypothesis for the study is needed.
Answer: Hypothesis for the study is incorporated suitably in the introduction.
- In results section, statistical data are needed. Please provide F value, degree of freedom and P-value obtained from the analysis of variance.
Answer: The earlier data was assessed by Duncan’s multiple range test. As suggested by reviewer entire data was assessed by Tukey’s test and it has been present in the revised manuscript. F value, degree of freedom and P-value obtained from the analysis of variance are provide in the manuscript.
4.Some statistical analysis are missing. Please, revise the results in figures 8 and 11.
Answer: The results presented in figures 8 and 11 are re-presented with statistical analysis.
- Other revision could be checked in pdf attached.
Answer: All the corrections suggested in the annotated copy are incorporated in the revised manuscript.

Reviewer 2 Report
This manuscript studies the production of secondary metabolites from cell cultures of Sageretia thea (Osbeck) M.C. Johnst. using balloon-type bubble bioreactors. It is well designed, well written and provides some interesting scientific information.
Author Response
Response to reviewers’ comments
Manuscript ID: plants-2250568
Authors are thankful to anonymous reviewers for their valuable comments on the manuscript. We have revised the manuscript in the light of reviewer’s comments and incorporated all the suggestion made by them. All corrections made in the revised manuscript have been done with track changes. Following are the specific changes made in the manuscript.

Reviewer 3 Report
The reviewed paper is interesting. The Authors described with details establishment of callus cultures, cel suspension cultures, some experiments with optimisation of cultures growth and production of several bioactive compounds in extracts form Sageretia thea. Additionally, the Authors also described elicitation of cell suspension cultures and the experiments with balloon-type bubble bioreactor. The descriebd experiments are new, modern, well designed, performed and clearly described and well discussed. However, in my opinion the Authors should better underline the novelty of the investigations. They wrote about cell and organ cultures of other plant species, but there is no information in Introduction section about in vitro cultures of Sageretia thea.
Some minor remarks:
1. Line 36 It should be Rhamnaceae instead of Ramnace
2. Line 98 It should be turned instead of tuned
3. Line 99 culture should be changed into subculture
4. Figure 2 should be presented as Figure1, because presented explant types and morphology of callus.
5. Lines 405-407 The explanation of growth index calculation is not clear. According to literature data (Grzegorczyk, I.; WysokiÅ„ska, H. Antioxidant compounds in Salvia officinalis L. shoot and hairy root cultures in the nutrient sprinkle bioreactor. Acta Soc. Bot. Pol. 2010, 79, 7–10; Weremczuk-Jeżyna et al., Transformed shoots of Dracocephalum forrestii W.W. Smith from different bioreactor systems as a rich source of natural phenolic compound. Molecules 2020, 25, 4533; doi:10.3390/molecules25194533; Jafernik et al., Plant Cell, Tissue and Organ Culture (PCTOC) (2020) 143:45–60 https://doi.org/10.1007/s11240-020-01895-2) growth index can be calculated for fresh and dry weight, separately according to the formula: final biomass (fresh or dry) – initial biomass (fresh or dry)/initial biomass (fresh or dry).
6. Line 409 The Authors should explain what does „freeze-dried cells” mean? It is not clear for a reader. It should be explained.
To conclude, I recommend the paper for publication after minor revision.
Author Response
Response to reviewers’ comments
Manuscript ID: plants-2250568
Authors are thankful to anonymous reviewers for their valuable comments on the manuscript. We have revised the manuscript in the light of reviewer’s comments and incorporated all the suggestion made by them. All corrections made in the revised manuscript have been done with track changes. Following are the specific changes made in the manuscript.
Reviewer #3
- Line 36 it should be Rhamnaceae instead of Ramnace
Answer: Correction has been incorporated.
- Line 98 it should be turned instead tuned
Answer: Correction has been incorporated.
- Lin 99 culture should be changed into subculture
Answer: Correction has been incorporated.
- Figure 2 should be presented as Figure 1, because presented explant types and morphology of callus.
Answer: Morphology of callus obtained from different explants has been shown in Figure 2 whereas data with statistical treatment is presented in Figure 1.
- Lines 405-407 – The explanation of growth index calculation is not clear. …
Answer: The growth index is calculated by both on fresh weight and dry weight data basis. However, the data based dry weight basis in appropriate because, there might lot of variation with fresh weight data due water content.
In the revised manuscript explanation of growth index calculation is given clearly. Viz. The growth index (Gi) was calculated on the basis of weight of dry tissue according to formula: Gi = (DW1 − DW0)/DW0: where DW0 was the weight of inoculum and DW1 the final weight of tissue after culture growth period.
- Line 409 The authors should explain what does “freeze-dried cells” mean?
Answer: For the analysis bioactive compounds especially, oven dried biomass may lose certain bioactive compounds such as phenolic, flavonoids and others. Therefore, to overcome this problem normally certain samples of biomass (harvested cells) are freeze dried and freeze-dried cells are useful for the analysis of bioactive compounds.

Round 2
Reviewer 1 Report
The manuscript “Production of secondary metabolites from cell cultures of Sageretia thea (Osbeck) M.C. Johnst. using balloon-type bubble bioreactors” has been improved and all my questions were taken into account.
I recommend the publication in “Plants”.